# Sialokinin in mosquito saliva shifts human immune responses towards intracellular pathogens

Jennifer L. Spencer Clinton[1¤a‡], Megan B. Vogt[1,2¤b‡], Alexander R. Kneubehl[1], Brianne M. Hibl[3¤c], Silke Paust[4], Rebecca Rico-Hesse[1] *

**1** Department of Molecular Virology and Microbiology, Baylor College of Medicine, Houston, Texas, United States of America, **2** Integrative Molecular and Biomedical Sciences Graduate Program, Baylor College of Medicine, Houston, Texas, United States of America, **3** Center for Comparative Medicine, Baylor College of Medicine, Houston, Texas, United States of America, **4** Department of Immunology and Microbiology, Scripps Research Institute, La Jolla, California, United States of America

¤a Current address: Department of Pediatrics and Tropical Medicine, Baylor College of Medicine, Houston, Texas, United States of America
¤b Current address: Department of Biomedical Sciences and Pathobiology, Virginia Polytechnic Institute and State University, Blacksburg, Virginia, United States of America
¤c Current address: Laboratory Animal Care Unit, University of Tennessee Health Science Center, Memphis, Tennessee, United States of America
‡ These authors share first authorship on this work.
* rebecca.rico-hesse@bcm.edu

**Data Availability Statement:** All FACS data can be accessed at the following repository: https://flowrepository.org/id/RvFrAlEwvzsqsMG2jdlUvN

## Abstract

Mosquito saliva is a mix of numerous proteins that are injected into the skin while the mosquito searches for a blood meal. While mosquito saliva is known to be immunogenic, the salivary components driving these immune responses, as well as the types of immune responses that occur, are not well characterized. We investigated the effects of one potential immunomodulatory mosquito saliva protein, sialokinin, on the human immune response. We used flow cytometry to compare human immune cell populations between humanized mice bitten by sialokinin knockout mosquitoes or injected with sialokinin, and compared them to those bitten by wild-type mosquitoes, unbitten, or saline-injected control mice. Humanized mice received 4 mosquito bites or a single injection, were euthanized after 7 days, and skin, spleen, bone marrow, and blood were harvested for immune cell profiling. Our results show that bites from sialokinin knockout mosquitoes induced monocyte and macrophage populations in the skin, blood, bone marrow, and spleens, and primarily affected CD11c- cell populations. Other increased immune cells included plasmacytoid dendritic cells in the blood, natural killer cells in the skin and blood, and CD4+ T cells in all samples analyzed. Conversely, we observed that mice bitten with sialokinin knockout mosquitoes had decreased NKT cell populations in the skin, and fewer B cells in the blood, spleen, and bone marrow. Taken together, we demonstrated that sialokinin knockout saliva induces elements of a $T_H1$ cellular immune response, suggesting that the sialokinin peptide is inducing a $T_H2$ cellular immune response during wild-type mosquito biting. These findings are an important step towards understanding how mosquito saliva modulates the human immune system and which components of saliva may be critical for arboviral infection. By

ZyrNVPaq4yiVTrGzeJgCfcpwdKgsVFWRFwSsFszego.

**Funding:** This work was supported by a National Institutes of Health grant to RRH (R01 A1099483). This project was supported by the Cytometry and Cell Sorting Core at Baylor College of Medicine and the expert assistance of Joel M. Sederstrom. The funders had no role in study design, data collection and analysis, decision to publish, or preparation of the manuscript.

**Competing interests:** The authors have declared that no competing interests exist.

identifying immunomodulatory salivary proteins, such as sialokinin, we can develop vaccines against mosquito saliva components and direct efforts towards blocking arboviral infections.

## Author summary

Numerous studies have shown the effects of mosquito saliva proteins on the immune system of animals and humans with disease caused by mosquito-borne pathogens. We have previously described some of these effects in humanized mice (which contain specific human immune system cells and develop arboviral diseases similar to humans) infected by mosquito bite with dengue and chikungunya viruses. In this study, we show that humanized mice have altered cellular immune responses after they are bitten by uninfected mosquitoes lacking the sialokinin salivary protein. Our results suggest that sialokinin alone shifts mammalian immunity towards a $T_H2$ response, away from the anti-viral, cell-mediated, and humoral responses that would protect against viruses included in the saliva. This is the first study of its kind, and it highlights how the effects of specific saliva components can be evaluated for human therapeutic intervention.

## Introduction

Mosquitoes can transmit arboviral or parasitic infections to human hosts, including dengue virus (DENV), West Nile virus (WNV), Zika virus (ZIKV), chikungunya virus (CHIKV), and malaria [1]. Currently, these infections account for approximately 750,000 yearly deaths, but incidences are expected to increase because of global warming and altered mosquito ranges [2–4]. It has been well documented that the mosquito vector itself influences the severity of mosquito-transmitted diseases, and that mosquito saliva itself plays a pivotal role in disease severity [5]. Mosquito saliva is a mixture of proteins that facilitate mosquito feeding by limiting platelet aggregation, coagulation, and pain responses, while promoting vasodilation and inflammation after a mosquito's blood meal [6]. Previous data on mosquito-borne viral infections have demonstrated that mice bitten by virus-infected mosquitoes, or needle inoculated with virus supplemented with mosquito saliva, have more severe disease than mice that were infected by needle injection of viruses alone [5,7–11]. Similarly, parasitic infections supplemented with mosquito saliva show increased disease severity, infectivity, and progression [12,13]. Importantly, there is little known about the mechanisms driving mosquito saliva enhancement of mosquito-borne viral pathogenesis. It has been hypothesized that mosquito saliva enhances viral pathogenicity by altering host immune responses. This is well supported by data indicating that mosquito saliva proteins are immunogenic to humans and can cause severe allergic reactions [14,15]. We have shown that mosquito saliva influences immune responses while studying human cellular immune responses in humanized mice bitten by uninfected *Aedes aegypti* mosquitoes and human peripheral blood mononuclear cells treated with mosquito saliva [16]. However, studies of human immunity to mosquito saliva proteins have been limited, mainly due to the lack of purified and safe testing reagents.

The two main components of the human immune system are the innate and adaptive immune responses. The innate immune response first recognizes a pathogen and triggers release of cytokines/chemokines to recruit innate immune cells, including monocytes, macrophages, dendritic cells (DC), neutrophils, and natural killer (NK) cells. These cells can directly

kill pathogens, or process and present antigens to stimulate T or B cell (adaptive immune cells) activation. Canonically, T cells are categorized by expression of either CD4 or CD8 molecules [17]. However, double positive T cells (DPT) express both CD4 and CD8 and have similar functions to CD8 T cells [18–21]. While CD8 T cells can directly kill viral-infected cells as cytotoxic T lymphocytes (CTLs), CD4-expressing T cells are able to differentiate into different subsets of T helper cells (denoted as $T_H1$ or $T_H2$) with differing functions [17,22,23]. A $T_H1$ response, or anti-viral, intracellular pathogen response, can be triggered by IL-12 or IL-18 secretion from DCs or NK cells in response to an intracellular pathogen. Conversely, a $T_H2$ response, or extracellular pathogen response, is triggered by release of IL-4 by DCs, mast cells, or NKT cells. Subsequently, $T_H1$ CD4 T cells stimulate CTL activation or B cell IgG production via TNFβ, IL-2, and IFNγ secretion [24]. $T_H2$ CD4 T cells release IL-4, IL-5, IL-6, IL-9, IL-10, and IL-13 to promote immune cell survival and limit phagocytic inflammation. Importantly, another subsets of immune cells, NKT cells, express both NK and T cell markers and can secrete cytokines to drive a $T_H1$ or $T_H2$ response (IFNγ, IL-4, IL-17A) [25–27]. While a dual $T_H1/ T_H2$ response can occur in the context of a single infection, $T_H1$ and $T_H2$ cytokines actively dampen cytokine production and immunity stimulated by the other response type [22,28,29]. Based on this knowledge of the immune system, it is hypothesized that mosquito saliva enhances viral pathogenesis by inducing a $T_H2$ (anti-parasitic) immune response, which inhibits crucial elements of the $T_H1$ (anti-viral) immune response.

Thus far, we have established that mosquito saliva does affect human immune cells in the humanized NOD scid gamma (hu-NSG) mouse model, resulting in a mixed $T_H1$ and $T_H2$ response [16]. Next, we wanted to determine which mosquito saliva proteins are responsible for inducing the immune responses that we previously observed. The first salivary protein that we investigated was the protein sialokinin, which is highly expressed in *Ae. aegypti* mosquito salivary glands [30]. Sialokinin is a 10 amino acid-long peptide that is cleaved from a pre-propeptide by unknown proteases [31,32]. Sialokinin functions as the only vasodilator specific to *Ae. aegypti* and serves to promote feeding success in the mosquito [30,31,33]. The structure and function of sialokinin are homologous to the mammalian tachykinin, Substance P [30,31]. Tachykinins are a group of neuropeptides that not only regulate contraction of smooth muscles and inflammation at mucous membranes but also mediate skin inflammation, including contact dermatitis, and pruritis [34]. Substance P and its primary receptor, neurokinin 1 receptor, are expressed in immune cells and promote inflammatory cytokine production by monocytes, survival of dendritic cells, suppression of NK cell numbers and activity, and proliferation of T cells [34,35].

Like its mammalian homologue, Substance P, sialokinin may also be immunomodulatory. When injected into C3H/HeJ mice, sialokinin decreased production of $T_H1$ cytokines and increased production of $T_H2$ cytokines by mouse splenocytes [36]. Recent studies have also shown that sialokinin enhances arboviral infection by modulating endothelial barrier function and altering immune cell recruitment to the bite site [33,37]. However, these studies were performed in wild-type C57BL/6 and BALB/c mice and gave limited insight on human immune responses. Based on this information, we hypothesized that sialokinin may also induce a $T_H2$ immune response in humans, thus shifting immune responses away from a protective antiviral response. To test this, we used uninfected sialokinin-knockout *Ae. aegypti* mosquitoes to bite humanized mice. If sialokinin does induce a $T_H2$ response, then saliva from sialokinin knockout mosquitoes should induce a higher $T_H1$ response than bites from wild-type mosquitoes. In addition to investigating the initiation of $T_H1$ and $T_H2$ immune responses, this study also assessed the recruitment of arbovirus-susceptible cells to the skin and the effects of sialokinin on immune cell populations.

## Materials & methods

### Ethics statement

All experiments involving mice were done in accordance with guidelines of the Institutional Animal Care and Use Committee at Baylor College of Medicine (IACUC Protocol AN-6151), and the recommendations in the *Guide for the Care and Use of Laboratory Animals* (Institute for Laboratory Animal Research, National Research Council, National Academy of Sciences, 2011).

### Mosquito rearing

*Ae. aegypti* (Rockefeller strain) mosquitoes were obtained from BEI resources as eggs (MRA-734). Mosquitoes were maintained under standard insectary conditions (~28°C, 80% relative humidity) with a 12-hour light/dark cycle maintained by the Philips Hue Smart Lighting system. Larvae were raised in water pans and fed on a mixture of ground rabbit chow (Purina)-liver powder (Bio-Serv)-yeast (Bio Serv) in a 4:1:1 ratio, *ad libitum*. Emerged mosquitoes were moved to mesh cages and fed on 10% sucrose (Sigma) solution *ad libitum*. Colony maintenance was performed by feeding mosquitoes on anesthetized C57/B6 mice. In the subsequent days following blood feeding, eggs were collected, desiccated, and stored for a maximum of 6 months.

Two sialokinin knockout *Ae. aegypti* (Liverpool) mosquito lines were created by using the CRISPR/Cas9 system and donated by the Adelman and Calvo laboratories [37]. The sialokinin d5 knockout line has a 5 base pair insertion, while the knockout line sialokinin d8 has an 8 base pair deletion. Both knockout lines were homozygous and contain frameshift mutations that resulted in premature stop codons. Sialokinin knockouts were verified via mass spectrometry and immunofluorescence for the presence of sialokinin in the saliva and salivary glands, respectively [37]. Our laboratory received sialokinin knockout mosquito lines as eggs and were reared in the same manner as wild-type mosquitoes.

### Production of Hu-NSG mice

Humanized mice were engrafted as previously described [9]. Briefly, male and female NSG breeders were obtained from The Jackson Laboratory (Bar Harbor, ME), and mice were bred in the Transgenic Mouse Facility at Baylor College of Medicine. The number of mice per group and sex of mice per group are described in **Table 1**. One day post-birth, each pup from these breedings was sublethally irradiated with 100 centigrays and intrahepatically injected with $3\times10^5$ CD34+ stem cells. These stem cells were isolated from human umbilical vein cord blood from the University of Texas MD Anderson Cord Blood Bank (Houston, TX) using the Dynabeads CD34 positive selection kit (Invitrogen) following the manufacturer's instructions. Levels of engraftment of human hematopoietic cells were tested 6 to 8 weeks later using flow

**Table 1. Hu-NSG mice used in this study.**

| Treatment Group | Sample size (n) | Number of males/females |
| --- | --- | --- |
| Wild-type mosquito | 12 | 7/5 |
| Unbitten Control | 6 | 5/1 |
| d5 mosquito | 6 | 4/2 |
| d8 mosquito | 7 | 4/3 |
| Sialokinin injection | 6 | 2/4 |
| Saline Injection | 6 | 3/3 |

cytometry to target human and mouse CD45+ cells. Mice that were at least 10% engrafted were used in this study. Typical mouse engraftment levels ranged from 15–75%.

## Mosquito biting and injection of Hu-NSG mice

Mosquito biting of reconstituted humanized mice was carried out as previously reported [9], although with uninfected wild-type or sialokinin knockout mosquitoes. In short, 4 to 7 days post-emergence, female mosquitoes were starved for 24 hours in dram vials (4–6 mosquitoes per vial) capped in a fine, white polyester mesh (Bio-Serv). Dram vials were kept at insectary conditions (28°C, 80% humidity) for the duration of the 24-hour starving. Mosquitoes were then transferred to a biosafety level-3 (BSL-3) facility, and the dram vials were held against a footpad of anesthetized, humanized mice, allowing the mosquitoes to feed. A "bite" was defined visually by mosquito engorgement and did not include probing; approximately 4 bites total occurred for each mouse and were distributed across both footpads. This number was chosen based on our previous studies demonstrating that 4 infected mosquitoes are required to bite each humanized mouse to consistently produce dengue fever [9].

Purified sialokinin peptide (BioBasic Inc) was synthesized as the mature peptide with a C-terminal amidation [38]. Sialokinin or saline were injected into Hu-NSG mice as additional controls. Hu-NSG mice were anesthetized with isoflurane prior to injection. Insulin syringes were used to deposit liquid into the rear footpad of the mice. Hu-NSG mice received 100μl of 36 μg/mL (3600ng per injection) of sialokinin diluted in sterile saline or an equivalent volume of sterile saline. The concentration of sialokinin per injection was chosen based on the amount of sialokinin present in salivary glands. Studies have shown that there is roughly 700ng of sialokinin I in a pair of *Ae. aegypti* salivary glands [31,39]. Since bitten mice received four mosquito bites by wild-type or knockout mosquitoes for the comparative studies, we multiplied the concentration of sialokinin by four and added approximately 20% to account for discrepancies in pipetting, mixing, and injection.

## Tissue collection and processing

Seven days post mosquito bite, mice were humanely euthanized via isoflurane overdose. Upon cessation of breathing, mice were exsanguinated via intracardiac bleed using a 25G 3/8-inch long needle. Blood was stored in heparin-treated microcentrifuge tubes for further processing. Skin from rear footpads was removed using surgical scissors and stored separately in PBS/FBS and 5μg/mL collagenase. Spleens and femurs were also removed from each mouse and stored separately in PBS/FBS.

Blood stored in heparinized tubes was transferred to 50mL conical tubes. Red blood cells were lysed using RBC lysis solution (eBioscience) according to the manufacturer's protocol. The remaining white blood cell pellet was resuspended at $1x10^4$ to $1x10^6$ cells/mL in PBS/2% FBS. These cells were stored at 4°C until stained for flow cytometry analysis.

Skin from footpads were cut into small pieces and incubated in PBS/FBS and 5mg/mL collagenase at 37°C for 1 hour. Following digestion, skin pieces were ground over a 40μm cell strainer into a 50mL conical tube. Skin cells were washed twice in PBS/FBS and resuspended at $1x10^4$ to $1x10^6$ cells/mL in PBS/FBS.

Spleens were burst by grinding between two frosted microscope slides. Spleen contents were then ground over a 40μm strainer into a 50mL conical tube. Red blood cells were lysed using RBC lysis solution (eBioscience) according to the manufacturer's protocol. Remaining cells were resuspended at $1x10^6$ to $1x10^7$ cells/mL in PBS/FBS.

Bone marrow was flushed out of femurs using a 25G needle filled with PBS/FBS. Marrow was ground over a 40μm strainer into a 50mL conical tube. Red blood cells were lysed and remaining cells were resuspended as with the spleen cells.

### Flow cytometry

Blood, bone marrow, skin, and spleen cells from hu-NSG mice were transferred to 96-well plates and incubated with antibodies against extracellular targets (**Table 2**) on ice for 30 minutes. Cells were fixed and permeabilized using the FoxP3 Transcription Factor Staining Buffer Kit (eBioscience) following the manufacturer's protocol. Following permeabilization, cells from hu-NSG mice were incubated with antibodies against intracellular targets (**Table 2**) on ice for 30 minutes. Cells were washed, resuspended in PBS/FBS, and stored at 4˚C until analysis. Samples were analyzed on the LSRII Fortessa (BD) using the HTS module. Data were collected using the FACSDiva software (BD) and analyzed using FlowJo (v10.2; FlowJo, LLC). Flow cytometry gating strategies used to differentiate immune cell populations are described in S1 Fig.

### Statistical analyses

Statistical analysis was performed using Prism (v6.0; GraphPad) software. Outliers were removed using ROUT analysis (Q = 1%). Data were analyzed via two-way ANOVA and t-tests using Holm-Sidak correction for multiple comparisons.

## Results

### Sialokinin recruits arbovirus-susceptible innate immune cells to the skin

To determine whether sialokinin modulates the human immune system, we subjected hu-NSG mice to 4 bites each from 2 lines of sialokinin knockout mosquitoes (denoted as d5 and d8). At 7 days post-bite, mice were euthanized and tissues were collected. Changes in immune cell populations were determined via flow cytometry. Data were compared to previous

**Table 2. Antibodies Used in Flow Cytometry Analysis of Tissues from Hu-NSG Mice Bitten by *Ae. aegypti* Mosquitoes.**

| Target | Target Location[a] | Panel(s) | Clone | Fluorophore | Manufacturer |
|---|---|---|---|---|---|
| CD3 | EC | 1, 2 | UCHT1 | BUV661 | BD Biosciences |
| CD4 | EC | 1 | OKT4 | BV650 | BioLegend |
| CD8a | EC | 1 | RPA-T8 | BV605 | BioLegend |
| CD11b | EC | 2 | ICRF44 | BV605 | BioLegend |
| CD11c | EC | 2 | 3.9 | BV650 | BioLegend |
| CD14 | EC | 2 | HCD14 | AF700 | BioLegend |
| CD19 | EC | 2 | HIB19 | PE/Cy7 | BioLegend |
| CD45 | EC | 1, 2 | 2D1 | Amcyan | BD Biosciences |
| CD56 | EC | 1 | HCD-56 | BV421 | BioLegend |
| CD80 | EC | 2 | 2D10 | APC | BD Biosciences |
| CD86 | EC | 2 | IT2.2 | PE/Cy5 | BD Biosciences |
| CD123 | EC | 2 | 7G3 | PE/CF594 | BD Biosciences |
| FoxP3 | N | 1 | 150D | AF647 | BioLegend |
| HLA-DR | EC | 2 | L243(G46-6) | APC/Cy7 | BD Biosciences |
| Ki67 | N | 2 | B56 | BV786 | BD Biosciences |

[a] EC = extracellular; IC = intracellular; N = nuclear

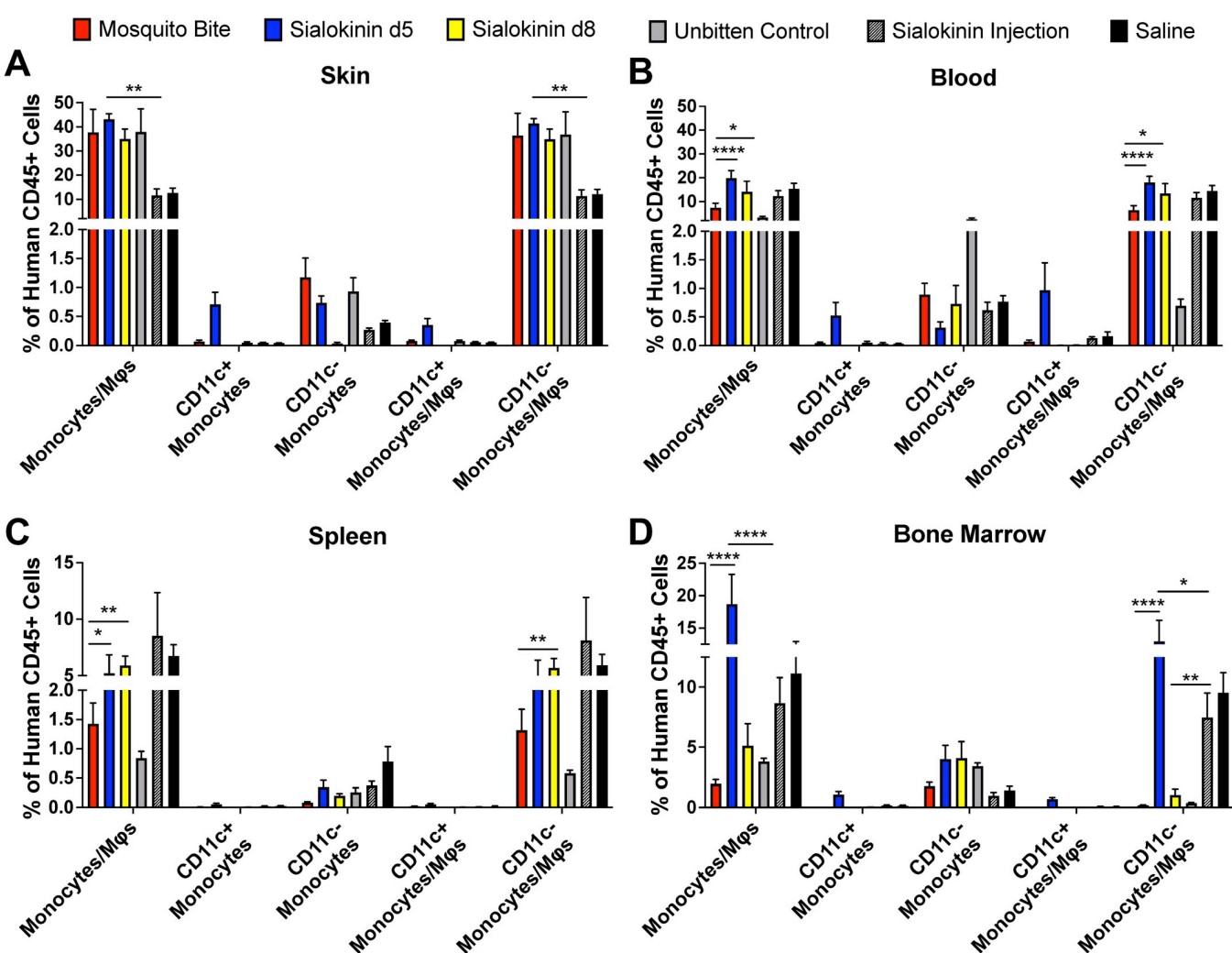

**Fig 1. macrophage populations in humanized mice/macrophage populations in humanized mice.** Hu-NSG mice were subjected to four bites each from sialokinin knockout mosquitoes and were euthanized 7 days postbite. Skin (**A**), blood (**B**), spleen (**C**), and bone marrow (**D**) were collected and changes in monocyte/macrophage populations were assessed via flow cytometry. Mosquito Bite indicates mice bitten by wild-type mosquitoes, and Unbitten indicates mice that were not bitten by mosquitoes. Data are represented as the mean percentage of the cell population out of total human CD45+ cells. Error bars represent 1 standard error of the mean (SEM). Statistical significance was determined using two-way ANOVA followed by multiple comparison t-tests using the Holm-Sidak correction. The threshold for significance was $p < 0.05$. Asterisks indicate: * = $p < 0.05$; ** = $p < 0.01$; **** = $p < 0.001$. Cell markers used to describe populations: Monocytes/Mφs: CD45+, CD3-, CD14+; CD11c+ Monocytes: CD45+, CD3-, CD11c+, CD11b+; CD11c- Monocytes: CD45+, CD3-, CD11c-, CD11b+; CD11c+ Monocytes/Mφs: CD45+, CD3-, CD14+, CD11c+; CD11c- Monocytes/Mφs: CD45+, CD3-, CD14+, CD11c-. Abbreviations: Mφ, Macrophage.

experiments in which mice were either unbitten or bitten by wild-type mosquitoes. Additional controls used for comparison were hu-NSG mice injected with purified sialokinin peptide alone or saline. Innate immune cell populations, such as total monocytes and macrophages (CD45+, CD3-, CD14+) were not significantly different in the skin, although sialokinin injection alone significantly decreased CD11c- monocytes and macrophages (CD45+, CD3-, CD11c-, CD11b+) compared to knockouts (**Fig 1A**). Conversely, CD11c- monocytes and macrophages were significantly increased in the blood, spleen, and bone marrow of sialokinin knockout-bitten mice, as compared to those bitten by wild-type mosquitoes (**Fig 1B, 1C and 1D**). Most of the monocytes and monocyte/macrophage populations observed are CD11c-, and there are no significant increases in CD11c+ populations. CD11c upregulation in monocytes and

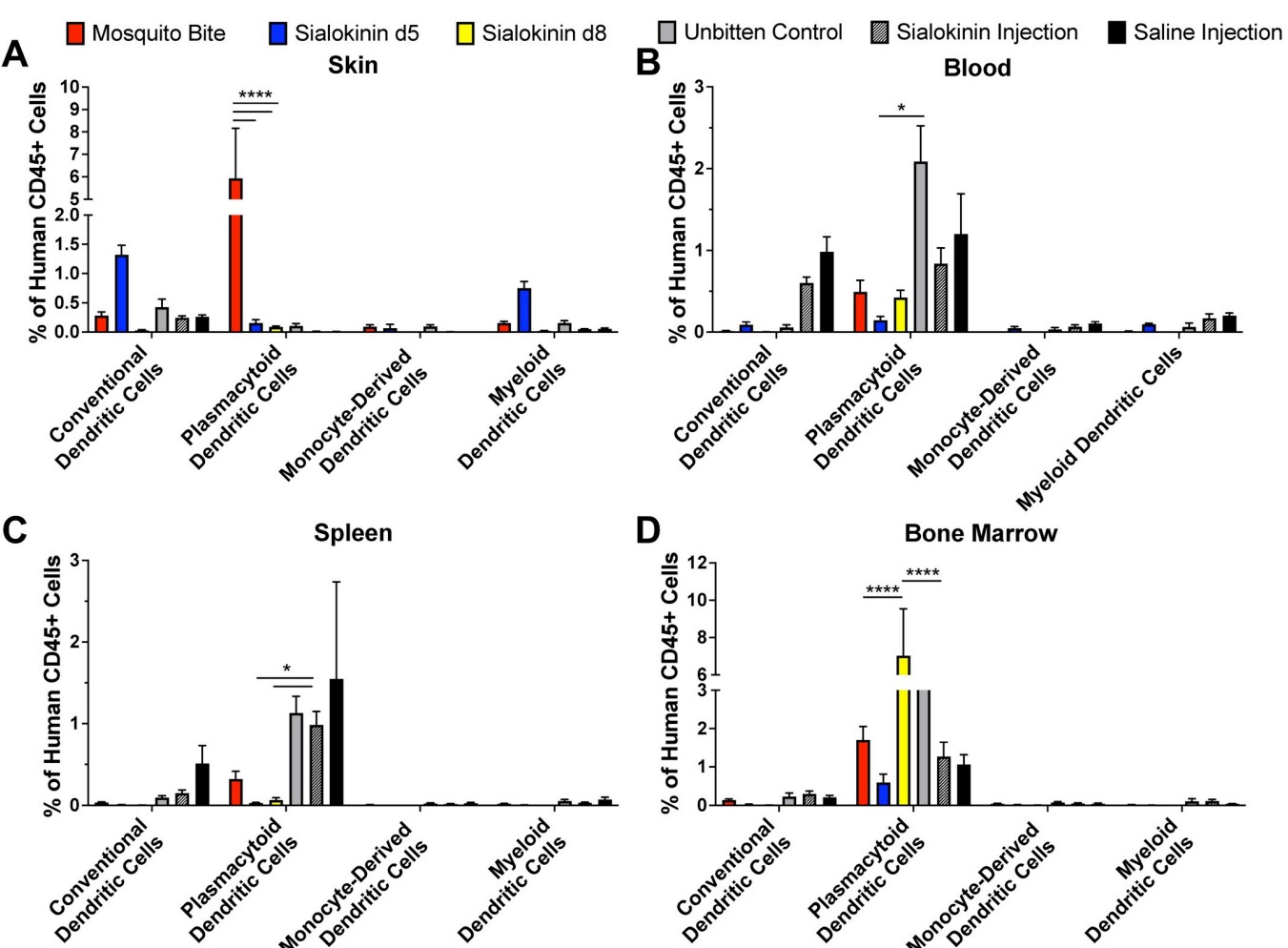

**Fig 2. Sialokinin knockout mosquito bites alter levels of plasmacytoid dendritic cells in humanized mice.** Hu-NSG mice were subjected to four bites each from sialokinin knockout mosquitoes and were euthanized 7 days postbite. Skin (**A**), blood (**B**), spleen (**C**), and bone marrow (**D**) were collected and changes in DC populations were assessed via flow cytometry. Mosquito Bite indicates mice bitten by wild-type mosquitoes, and Unbitten indicates mice that were not bitten by mosquitoes. Data are represented as the mean percentage of the cell population out of total human CD45+ cells. Error bars represent 1 SEM. Statistical significance was determined using two-way ANOVA followed by multiple comparison t-tests using the Holm-Sidak correction. The threshold for significance was $p < 0.05$. Asterisks indicate: * = $p < 0.05$; **** = $p < 0.001$. Cell markers used to describe populations: Conventional dendritic cells: CD3-, CD19-, CD14-, CD123-, CD11c+; Plasmacytoid dendritic cells: CD3-, CD19-, CD14-, CD123+, CD11c-; Monocyte derived dendritic cells: CD3-, CD19-, CD14-, CD11b-, CD11c+, HLA-DR+; myeloid dendritic cells: CD3-, CD19-, CD14-, CD11b+, CD11c+.

macrophages are associated with inflammatory conditions and can result in increased integrin adherence of CD11c+ cells in the blood vessels, ultimately leading to increased recruitment of those cells to sites of interest. Taken together, this data indicates that large amounts of inflammation did not occur.

Dendritic cells are another innate immune cell population that is a known target of arboviral infection. Thus, we also analyzed changes in multiple dendritic cell populations post-mosquito bite or injection. Hu-NSG mice bitten by sialokinin knockout mosquitoes showed significantly decreased populations of plasmacytoid dendritic cells (CD3-, CD19-, CD14-, CD123+, CD11c-) in the skin, as compared to wild-type bitten mice (**Fig 2A**). Plasmacytoid dendritic cells were significantly higher in the blood and spleen of sialokinin injected mice than that of mice bitten by sialokinin knockout mosquitoes (**Fig 2B and 2C**). Alternatively, plasmacytoid dendritic cells were significantly higher in the bone marrow of mice bitten by

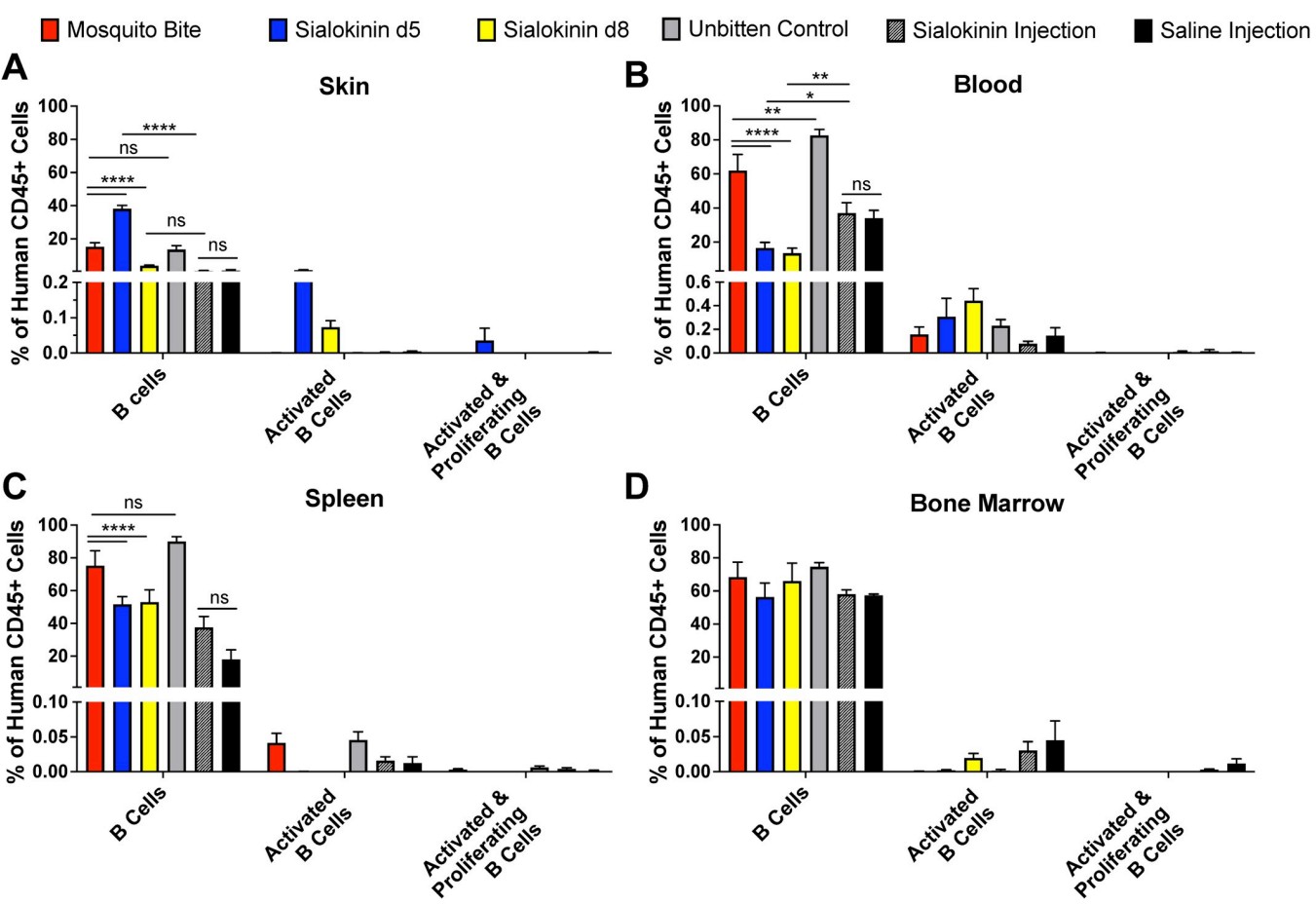

**Fig 3. Sialokinin knockout mosquito bites influence B cell populations in humanized mice.** Hu-NSG mice were subjected to four bites each from sialokinin knockout mosquitoes and were euthanized 7 days postbite. Skin (**A**), blood (**B**), spleen (**C**), and bone marrow (**D**) were collected and changes in B cell populations were assessed via flow cytometry. Mosquito Bite indicates mice bitten by wild-type mosquitoes, and Unbitten indicates mice that were not bitten by mosquitoes. Data are represented as the mean percentage of the cell population out of total human CD45+ cells. Error bars represent 1 SEM. Statistical significance was determined using two-way ANOVA followed by multiple comparison t-tests using the Holm-Sidak correction. The threshold for significance was $p < 0.05$. Asterisks indicate: * = $p < 0.05$; ** = $p < 0.01$; **** = $p < 0.001$. Cell markers used to describe populations: B cells: CD45+, CD3-, CD19+; Activated B cells: CD3-, CD19+, CD80+, CD86+; Activated and proliferating B cells: CD3-, CD19+, CD80+, CD86+, Ki67+. Abbreviations: ns = not significant.

sialokinin d8 knockout mosquitoes as compared to wild-type mosquito bite or sialokinin injection alone (**Fig 2D**). While we observed higher percentages of conventional dendritic cells (CD3-, CD19-, CD14-, CD123-, CD11c+) and myeloid dendritic cells (CD3-, CD19-, CD14-, CD11b+, CD11c+) in the skin of mice bitten by sialokinin d5 knockout mosquitoes, there were no significant differences (**Fig 2A**).

When assessing B cell populations, we observed decreased B cells (CD45+, CD3-, CD19+) in the blood and spleens of mice that were bitten by sialokinin knockout mosquitoes as compared to those that received bites from wild-type mosquitoes (**Fig 3B and 3C**). Inconsistent trends were observed in skin of sialokinin knockout bitten mice, with d5 increasing B cells and d8 decreasing B cells (**Fig 3A**). Additionally, few activated B cells (CD3-, CD19+, CD80+, CD86+) or activated and proliferating B cells (CD3-, CD19+, CD80+, CD86+, Ki67+) were observed in any group and across all tissues sampled (**Fig 3**). Furthermore, injection with the sialokinin peptide did not appear to fully rescue the effect of the knockout and resulted in similar or lower numbers of B cells than the other groups.

## Sialokinin induces elements of a $T_H2$ response

Lastly, T cell subsets and other lymphocyte populations were assessed after sialokinin knockout or wild-type mosquito bites. CD4 T cells (CD45+, CD3+, CD4+, CD8-) were significantly increased across all tissue types in mice bitten by the sialokinin d5 mosquitoes compared to the wild-type mosquitoes (**Fig 4A, 4C, 4E and 4G**). Additionally, CD8 T cells (CD45+, CD3+, CD4-, CD8+) were increased in the bone marrow of mice bitten by sialokinin d8 mosquitoes compared to those bitten by wild-type mosquitoes (**Fig 4G**). In the skin, NKT cells (CD45+, CD3+, CD56+) were significantly decreased in mice bitten by sialokinin knockout mosquitoes compared to mice bitten by wild-type mosquitoes (**Fig 4A**). Furthermore, NK cells were significantly increased in the skin and blood of mice bitten by sialokinin knockout mosquitoes compared to those bitten by wild-type mosquitoes (**Fig 4B**). As discussed previously, NKT cells and NK cells initiate $T_H2$ and $T_H1$ responses, respectively [28]. Thus, a decrease in NKT cells coincident with an increase in NK cells may indicate induction of a $T_H1$ response by sialokinin knockout saliva. Therefore, these data suggest that sialokinin induces elements of a $T_H2$ response. Lastly, we measured B cells as a percentage of total human leukocytes in harvested mouse tissues. B cells observed significant decreases in B cells in the blood, spleen, and bone marrow of mice bitten by sialokinin knockout mosquitoes compared to those bitten by wild-type mosquitoes (**Fig 4D, 4F and 4H**).

## Discussion

In addition to our previous investigations of the effects of mosquito saliva on the human immune system, we have now focused on which specific mosquito saliva proteins are immunomodulatory. We tested the effects of our first protein of interest, sialokinin, on the human immune system by subjecting hu-NSG mice to bites from sialokinin knockout mosquitoes. We observed that several immune cell populations were significantly different between humanized mice bitten by sialokinin knockouts mosquitoes, wild-type mosquitoes, or injected with sialokinin. These populations included CD11c- monocytes/macrophages, plasmacytoid dendritic cells, B cells, CD4+ T cells, CD8+ T cells, and NK cells. Taken together, our data show that sialokinin knockout saliva induces elements of a $T_H1$ response, suggesting that sialokinin induces elements of a $T_H2$ response.

Interestingly, we showed that sialokinin knockout mosquito bites yielded decreased plasmacytoid dendritic cells in the skin but increased their percentages in the bone marrow. Sialokinin injection also elicited an increase in plasmacytoid dendritic cells in the blood and spleen. This is consistent with plasmacytoid dendritic cell production in the bone marrow and high concentration in the blood; they are the most common type of dendritic cells in the blood [40]. Conversely, plasmacytoid dendritic cells have also been described in the skin during cases of psoriasis and skin inflammation [41,42], and they secrete large amounts of IFNα in response to infection [40]. However, it is currently unknown when plasmacytoid dendritic cells make their way to the skin, and skin translocation may occur immediately after mosquito bite. As IFNα is important in anti-viral response and is highly secreted by plasmacytoid dendritic cells, this may counter our hypothesis that mosquito saliva inhibits anti-viral immune responses.

Our data also indicate that B cells were decreased in the blood and spleens but had little or inconsistent effects in the skin and bone marrow. Interestingly, sialokinin injection did not seem to have a compensatory effect on B cell populations. As the sialokinin peptide used was only 10 amino acids in length and may be too small to stimulate a B cell response on its own. Additionally, the sialokinin peptide shares homology with mammalian Substance P (and other neurokinins). Therefore, B cells that may have responded to sialokinin probably underwent apoptosis or became anergic to avoid an anti-self-immune response. Sialokinin is the final

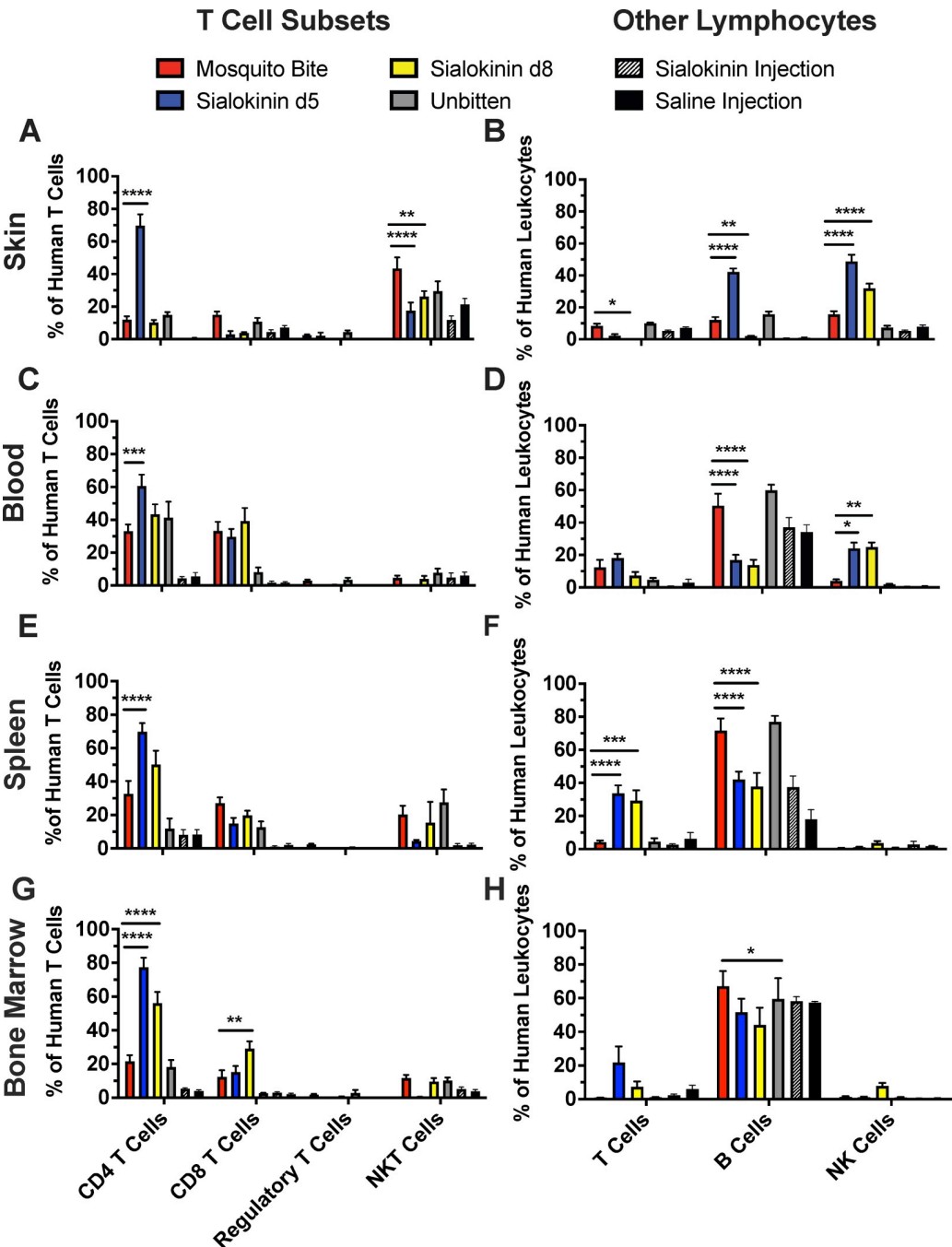

**Fig 4. Sialokinin knockout mosquito bites induce CD4+ T cell and NK cell populations in humanized mice.** Hu-NSG mice were subjected to four bites each from sialokinin knockout mosquitoes and were euthanized 7 days postbite. Skin, blood, spleen, and bone marrow were collected and changes in T cell (**A, C, E, G**) and other lymphocyte (**B, D, F, H**) populations were assessed via flow cytometry. Mosquito Bite indicates mice bitten by wild-type mosquitoes, and Unbitten indicates mice that were not bitten by mosquitoes. Data are represented as the mean percentage of the cell population out of total human T cell numbers or total human leukocytes. Regulatory T cell populations were not assessed in saline injection and sialokinin injection groups due to small percentages of CD4 T cells. Error bars represent 1 SEM. Statistical significance was determined using two-way ANOVA followed by multiple comparison t-tests using the Holm-Sidak correction. The threshold for significance was $p < 0.05$. Asterisks indicate: * = $p < 0.05$; ** = $p < 0.01$; *** = $p < 0.005$; **** = $p < 0.001$. Cell markers used to describe populations: CD4 T cells: CD45+, CD3+, CD4+, CD8-; CD8 T cells: CD45+, CD3+, CD4-, CD8+; Regulatory T cells: CD45+, CD3+, CD4+, CD8-, FoxP3+; NKT Cells: CD45+, CD3+, CD56+; T cells: CD45+, CD3+; B cells: CD45+, CD3-, CD19+; NK Cells: CD45+, CD3-, CD56+. Abbreviations: NKT, Natural K T Cell; NK, Natural Killer Cell.

product of cleavage of a pre-pro-peptide and whether the peptide cleavage occurs in the mosquito or in the host is currently unknown. If cleavage occurs in the host, then a B cell response may occur against the other portions of the pre-pro-peptide. Future studies with the sialokinin pre-pro-peptide could shed light on these differences. Furthermore, B cells may be associated with either $T_H1$ or $T_H2$ responses depending on what class of antibody they are actively producing [24]. We will not be able to determine with which immune response B cells are associated in our current model; human B cells in hu-NSG mice do not undergo class switching and are only able to produce IgM antibodies [43]. Future studies could incorporate the humanized-DRAG mouse model, which can produce all antibody classes [43,44].

By assessing changes in T cell subsets and lymphocytes, we described an increase in CD4 T cells in all tissue types evaluated and an increase in CD8 T cells in the bone marrow in sialokinin knockout mosquito bitten mice. CD4 T cells may be involved in propagating either $T_H1$ or $T_H2$ responses [22]. To determine if the increased CD4 T cells are associated with a $T_H1$ or $T_H2$ response, we would need to analyze cytokine production by these cells. IL-2 and IFNγ producing CD4 T cells would be associated with a $T_H1$ response, while IL-4 producing CD4 T cells would be associated with a $T_H2$ response. CD8 T cells are effector cells in the $T_H1$ response and kill virus-infected cells [17]. When found in the bone marrow, CD8 T cells are typically associated with immunological memory [45]. Based on our current flow cytometry panels, we were unable to determine whether the CD8 T cells found in the bone marrow are memory T cells. If mosquito saliva does alter immunological memory, it could alter the host's ability to respond quickly to viral infections. Our data also demonstrated decreased NKT cells in skin and increased NK cells in skin and blood of mice bitten by sialokinin knockout mosquitoes as compared to those bitten by wile-type mosquitoes. This result is consistent with previous studies investigating Substance P (the mammalian homologue of sialokinin), which determined that Substance P inhibits NK cell function and decreases NK cell populations [34].

While analyzing the results, we noted that mice bitten by sialokinin d5 mosquitoes did not always produce the same immune response as mice bitten by sialokinin d8 mosquitoes. These mosquito lines were created via the CRISPR/Cas9 system using two different guide RNAs; this would allow us to more easily identify off-target effects. We currently do not know if any other genes were affected during the process of knocking out sialokinin. It is possible that CRISPR/Cas9 knockout of sialokinin inadvertently introduced additional mutations that affected other saliva proteins, ultimately leading to different immune responses in the mice. To resolve this issue, we will need to further characterize the saliva of these mosquitoes, looking for changes in salivary protein composition between the two knockout lines and wild-type mosquitoes.

While injection with sialokinin was hypothesized to rescue phenotypes affected by sialokinin knockout mosquito bites, several cell populations did not show such an effect. While this is an interesting and unexpected result, it is important to consider the interplay between mosquito salivary proteins within the host. With this in mind, it is possible that the differences in "recovery" responses were due to injection of the sialokinin peptide alone and not the full breadth of mosquito salivary proteins present in the knockout bites. The lack of additional salivary proteins in the sialokinin-injected groups could be playing a role.

There have been relatively few studies of sialokinin's function in mosquito saliva, let alone with regards to the human immune response. Sialokinin itself was characterized in 1992 as a vasodilator in mosquitoes with functional similarities to tachykinins [30]. Later studies determined the nucleotide sequence of the gene encoding sialokinin and the amino acid sequence of the sialokinin peptide, which showed homology to a mammalian tachykinin, Substance P [31,32]. As tachykinins are known immunomodulatory proteins [34], we hypothesized that sialokinin would be immunomodulatory as well. Previous studies using the same sialokinin knockout mosquitoes as used in our study demonstrated that total leukocytes, neutrophils,

and CD8+ T cells were decreased in footpads of mosquito bitten BALB/c mice [37]. These data further support our findings that sialokinin has an immunomodulatory effect and alters immune cell recruitment. Additionally, a previous study of the effects of sialokinin on the host's immune system showed sialokinin injection into C3H/HeJ mice caused splenocytes to produce decreased amounts of $T_H1$ cytokines and increased amounts of $T_H2$ cytokines [36]; C3H/HeJ mice are susceptible to infection by mouse-adapted flaviviruses [46]. Interestingly, sialokinin did not alter cytokine expression in C3H/PRI-*Flv*$^r$ *(previously named C3H.Rv)*, a sub-line of C3H/HeJ mice that are resistant to flavivirus infection [36,47]. This result corroborates previous studies in humans and mice that correlate increased immune response, particularly allergic response, to mosquito saliva with an increased severity of arboviral disease [48–50]. Furthermore, these increased immune responses are not consistent among humans, potentially explaining why some individuals are more susceptible to severe arbovirus disease than others [49]. Whether sialokinin is immunogenic in only certain individuals or whether it specifically contributes to saliva-induced enhancement of viral disease remains unknown.

All of the mouse studies of mosquito saliva effects reported previously have used mice that differ from human immune systems: AG129 (which lack type I and type II IFN receptors), BALB/c mice (which are predisposed to $T_H2$ immune responses), C57BL/6 mice (which are predisposed to $T_H1$ immune responses), C3H/HeJ (which have a B cell deficiency), and C3H/PRI-*Flv*$^r$ (which are resistant to flavivirus infection) [36,37,47,48,51–54]. The humanized mouse model used in this study does not have the same limitations as previous models because these mice have been reconstituted with a human immune system, including innate immunity [55]. However, this model does not have a full human complement system or a fully functional T cell compartment [55]. In future studies, we seek to address these limitations by using other types of reconstituted humanized mice (e.g., BLT, DRAG), which would reproduce different parts of the human immune system (such as active T helper cells, immunoglobulin class-switching, etc.) following mosquito bite [44,56,57]. Additionally, we expect to test for the biological significance of these immune cell changes, which might lead to stimulation of infected cells to migrate to important sanctuary tissues (e.g., bone marrow or brain) where viral reservoirs could be established away from the full forces of the immune system. In the case of arboviruses, many establish infections in brain or bone marrow cells of human patients, leading to specific pathologies such as encephalitis, bone loss, leukopenia, and thrombocytopenia [58–60]. It is currently unclear if mosquito saliva contributes to these tissue infections and more severe pathologies.

Mosquitoes and the diseases they transmit are of growing public health concern. Often, there are no prophylaxes for these diseases other than mosquito control and no treatments other than palliative care. Understanding how mosquito saliva interacts with the human immune system not only helps us understand mechanisms of disease pathogenesis but also could provide possibilities for treatments. If we know which mosquito saliva components enhance pathogenesis of diseases, we could create a human vaccine to counteract these effects for multiple arbovirus infections. A similar approach has been used to vaccinate and protect mice against a sandfly saliva protein (maxadilan) that enhances the infection and progression of Leishmania major [61]. These approaches have also been commercialized and used to interrupt tick transmission of cattle diseases [62], and we expect that the definition of these factors would help provide the same approaches in humans.

## Supporting information

**S1 Fig. Representative flow cytometric gating strategy plots used here.**
(TIFF)

## Author Contributions

**Conceptualization:** Silke Paust, Rebecca Rico-Hesse.

**Data curation:** Jennifer L. Spencer Clinton, Megan B. Vogt, Alexander R. Kneubehl, Brianne M. Hibl.

**Formal analysis:** Megan B. Vogt.

**Funding acquisition:** Rebecca Rico-Hesse.

**Investigation:** Jennifer L. Spencer Clinton, Megan B. Vogt, Alexander R. Kneubehl, Brianne M. Hibl.

**Methodology:** Silke Paust, Rebecca Rico-Hesse.

**Project administration:** Rebecca Rico-Hesse.

**Resources:** Rebecca Rico-Hesse.

**Supervision:** Rebecca Rico-Hesse.

**Validation:** Jennifer L. Spencer Clinton, Megan B. Vogt.

**Visualization:** Jennifer L. Spencer Clinton, Megan B. Vogt.

**Writing – original draft:** Jennifer L. Spencer Clinton, Megan B. Vogt.

**Writing – review & editing:** Jennifer L. Spencer Clinton, Megan B. Vogt, Alexander R. Kneubehl, Brianne M. Hibl, Rebecca Rico-Hesse.

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
