## [Decision Letter · Decision Letter 0]

7 Oct 2022

Dear Dr. Rico-Hesse,

Thank you very much for submitting your manuscript "Sialokinin in mosquito saliva shifts human immune responses towards intracellular pathogens" for consideration at PLOS Neglected Tropical Diseases. As with all papers reviewed by the journal, your manuscript was reviewed by members of the editorial board and by several independent reviewers. The reviewers appreciated the attention to an important topic. Based on the reviews, we are likely to accept this manuscript for publication, providing that you modify the manuscript according to the review recommendations. 

Your manuscript has been evaluated by three reviewers, all of which consider this interesting work that will add to our understanding of mosquito-borne disease and vector biology. The reviewers did ask for clarification or minor editing on a number of points and we ask that you evaluate and respond to these comments and make appropriate changes to your manuscript.

Sincerely,

Richard A. Bowen

Academic Editor

Elvina Viennet

Section Editor

Your manuscript has been evaluated by three reviewers, all of which consider this interesting work that will add to our understanding of mosquito-borne disease and vector biology. The reviewers did ask for clarification or minor editing on a number of points and we ask that you evaluate and respond to these comments and make appropriate changes to your manuscript.

Reviewer's Responses to Questions

**Key Review Criteria Required for Acceptance?**

**Methods**

-Are the objectives of the study clearly articulated with a clear testable hypothesis stated?

-Is the study design appropriate to address the stated objectives?

-Is the population clearly described and appropriate for the hypothesis being tested?

-Is the sample size sufficient to ensure adequate power to address the hypothesis being tested?

-Were correct statistical analysis used to support conclusions?

-Are there concerns about ethical or regulatory requirements being met?

Reviewer #1: All of my comments appear in the section entitled "Summary and General Comments"

Reviewer #2: (No Response)

Reviewer #3: (No Response)

**Results**

-Does the analysis presented match the analysis plan?

-Are the results clearly and completely presented?

-Are the figures (Tables, Images) of sufficient quality for clarity?

Reviewer #1: All of my comments appear in the section entitled "Summary and General Comments"

Reviewer #2: (No Response)

Reviewer #3: (No Response)

**Conclusions**

-Are the conclusions supported by the data presented?

-Are the limitations of analysis clearly described?

-Do the authors discuss how these data can be helpful to advance our understanding of the topic under study?

-Is public health relevance addressed?

Reviewer #1: All of my comments appear in the section entitled "Summary and General Comments"

Reviewer #2: (No Response)

Reviewer #3: (No Response)

**Editorial and Data Presentation Modifications?**

Reviewer #1: (No Response)

Reviewer #2: (No Response)

Reviewer #3: (No Response)

**Summary and General Comments**

Reviewer #1: This is a well-conducted and nicely written study. The objective was to determine the effect of sialokinin (a potential immunomodulatory mosquito saliva protein) on the human immune response through the use of humanized mice. The mice were fed on by wildtype mosquitoes or sialokinin knockout mosquitoes (two strains were used) or they received sialokinin via needle inoculation. All of my concerns are extremely minor. 

Line 77: “or bitten by uninfected mosquitoes” – surely this is a mistake because disease would not be expected in mice bitten by these mosquitoes

Line 89: replace “include” with “are the”

Line 213: considering mentioning the % of FBS the first time “PBS/FBS” is used

Line 228: replace Table with Table 2. Same applies to line 231

Lines 253-254: I can see the CD11c+ data in Figure 1, but not the data for CD45+, CD3- and CD14+. Consider mentioning “data not shown” where applicable. Same applies to later figures.

Reviewer #2: The manuscript “Sialokinin in mosquito saliva shifts human immune responses towards intracellular pathogens” explores the potential immunomodulatory activity of sialokinin using humanized mice, which is of great importance to the field. The manuscript is well ordered and easy to understand. 

Overall, my main concern is the significant difference in results that can be observed between the two KO strains. Additionally, the injection of sialokinin, in most cases, does not “recover” the observed phenotype. Therefore, is not clear to me how the authors reached some of their conclusions.

Below I present a few specific questions to the authors.

1. In lines 200 – 201, the authors state that 18 ug of sialokinin were injected in each animal. What was the rationale for this concentration? Did the authors perform a pilot experiment using different concentrations of sialokinin? Considering that mosquito salivary glands contain 1 – 2 ug of total protein and it is speculated that not all salivary content is injected during the feeding cycle, it is more likely that only a few nano-grams of sialokinin are injected into the host.

2. Most results showed that the injection of sialokinin did not “recover” the observed phenotype when wild-type mosquitoes were used. One example of this is provided in figure 1C, in which an increase of CD11c- monocytes and macrophages were found when comparing KO and wild-type groups. However, the same effect is found in the groups injected with sialokinin or saline. If the increase of CD11c- population is due to the lack of sialokinin in the mosquito saliva, how did the injection of sialokinin yield the same result?

3. Some of the results presented in the current manuscript show inconsistencies between the two KO strains. One example is in lines 287 – 289, in which the authors state that plasmacytoid dendritic cells were higher in the bone marrow of mice bitten by sialokinin d8 KO, but an opposite effect was observed for the d5 KO (Figure 2D). These “contradictory” results can also be observed in other panels of figures 2, 3 and 4. Could the authors provide an explanation or a comment on this? Could the KO of sialokinin result in additional changes in the composition of mosquito saliva?

4. Data for the sialokinin- and saline- injected groups are missing in figure 4.

Reviewer #3: In the current study, Clinton et al., studied the role of Sialokinin; a mosquito saliva protein, in immune responses regulations following the mosquito bite. Although the Sialokinin has already been studied in details however majority of these studies are performed on wild-type C57BL/6 and BALB/c mice hence there is a need for studies which can insight more about molecular cross talk and immune responses. In the current study, authors used humanized mice and studied immune cells regulation following the bite with sialokinin knockout or wild type Ae aegypti. Overall this is nice study however I have few concern authors need to address-

1. Authors need to validate if mosquitoes are sialokinin knockout using western blotting .

2. Authors also need to check if addition of mosquito saliva or salivary gland extract alone to human immune cells like THP1 can alter the immune responses.

3. Please incorporate FACS graphs to the result section.

PLOS authors have the option to publish the peer review history of their article (what does this mean?). If published, this will include your full peer review and any attached files.

Reviewer #1: No

Reviewer #2: No

Reviewer #3: Yes: VIPIN SINGH RANA

Figure Files:

Data Requirements:

Reproducibility:

References

---

## [Editor Report · Decision Letter 1]

11 Jan 2023

Dear Dr. Rico-Hesse,

We are pleased to inform you that your manuscript 'Sialokinin in mosquito saliva shifts human immune responses towards intracellular pathogens' has been provisionally accepted for publication in PLOS Neglected Tropical Diseases.

Best regards,

Richard A. Bowen

Academic Editor

Elvina Viennet

Section Editor

---

## [Editor Report · Acceptance letter]

13 Jan 2023

Dear Dr. Rico-Hesse,

We are delighted to inform you that your manuscript, "Sialokinin in mosquito saliva shifts human immune responses towards intracellular pathogens," has been formally accepted for publication in PLOS Neglected Tropical Diseases.

Best regards,

Shaden Kamhawi

co-Editor-in-Chief

Paul Brindley

co-Editor-in-Chief
